# A Novel Endodontic Approach in Removing Smear Layer Using Nano and Submicron Diamonds with Intracanal Oscillation Irrigation

**DOI:** 10.3390/nano13101646

**Published:** 2023-05-15

**Authors:** Ching-Shuan Huang, Chih-Hsun Hsiao, Yu-Chia Chang, Chih-Hsiang Chang, Jen-Chang Yang, James L. Gutmann, Huan-Cheng Chang, Haw-Ming Huang, Sung-Chih Hsieh

**Affiliations:** 1School of Dentistry, College of Oral Medicine, Taipei Medical University, Taipei 11031, Taiwan; 2Department of Dentistry, Taipei Medical University Hospital, Taipei 11031, Taiwan; 3Department of Endodontology, Taipei Municipal Wan-Fang Hospital, Taipei 11696, Taiwan; 4Graduate Institute of Nanomedicine and Medical Engineering, College of Biomedical Engineering, Taipei Medical University, Taipei 11031, Taiwan; 5College of Dentistry, Texas A&M University, Dallas, TX 77843, USA; 6Institute of Atomic and Molecular Sciences, Academia Sinica Taiwan, Taipei 11529, Taiwan

**Keywords:** nano diamonds, submicron diamonds, EDTA, smear layer, endodontic irrigation

## Abstract

Sodium hypochlorite (NaOCl) and ethylenediaminetetraacetic acid (EDTA) are commonly recommended for effectively removing organic and inorganic components in the smear layer. This layer is found on root canal walls after root canal instrumentation. However, high-concentration EDTA reduces the strength of dentin and the dissolution efficacy of organic substances in NaOCl solution. The objective of this study was to investigate the efficacy of applying nano and submicron diamonds in irrigation solutions with sonic and ultrasonic oscillation for removing the smear layer during endodontic treatment. Extracted single-rooted human teeth were instrumented with ProTaper^®^ Gold (Dentsply Sirona) nickel–titanium rotary instruments. Subsequently, each canal was irrigated with 3% NaOCl, 17% EDTA, distilled water, and 10–1000 nm-sized nano and submicron diamond irrigation solutions, respectively. Sonic and ultrasonic instruments were compared for oscillating the irrigation solutions. The teeth were processed for scanning electron microscopy to observe the efficiency of smear layer removal on the canal walls. Our results indicated that diamond sizes of 50 nm and above irrigation solutions showed significant effectiveness in removing the smear layer following the oscillation of sonic instruments for 10 s. Ultrasonic assisted 500 nm and 1000 nm diamond solutions significantly differed from the other diamond-sized solution in their ability to remove the smear layer. These results suggest that sonic and ultrasonic oscillation with specific sizes of nano and submicron diamond irrigation solution can be used as an alternative approach to removing the smear layer during endodontic treatment. The potential clinical application of root canal treatments can be expected.

## 1. Introduction

The smear layer, composed of organic and inorganic materials, represents debris generated during the process of enlarging and shaping root canal walls when endodontic treatment is performed. Following the process, the root canal must be obturated to prevent the penetration of microorganisms and bacterial toxins into the prepared root canal. However, whether the presence of a smear layer on the surface of root canals affects the sealing ability is controversial [1,2,3,4,5]. Some studies reported that endodontic success rates increase with the presence of a smear layer since the smear layer can obstruct dentinal tubules, preventing bacterial exchange [6,7]. On the other hand, some advocated that the smear layer can nourish bacteria and cause the proliferation of microorganisms [5,8,9,10,11]. In addition, the smear layer acts as a barrier that hinders the penetration of intracanal medicaments into dentinal tubules [12,13,14,15,16,17]. Improved retention and better sealing ability are demonstrated with the mechanical interlocking of the sealer plug in dentinal tubules [18,19] with the removal of the smear layer. Despite the controversy over retaining or removal of the smear layer, complete debridement of the smear layer is crucial for successful root canal treatment [20].

The most common solution for removing the inorganic component of the smear layer is EDTA, which refers to the chelating agent with the formula [CH_2_N(CH_2_CO_2_H)_2_]_2_. This amino acid is frequently utilized to bind to di- and trivalent metal ions. EDTA binds to metals through four carboxylate and two amine groups. It undergoes a chelating reaction by reacting with the calcium ions in dentin and forming soluble calcium chelates. Although it can effectively remove the smear layer produced during canal shaping, several studies have reported negative effects of EDTA, being a calcium-depleting irrigant, on the radicular dentin beneath the smear layer. Applying 17% EDTA in root canals for more than 1 min can cause dentinal erosion [21,22]. Furthermore, EDTA could cause demineralizing effects on root canal dentin, resulting in a reduction in dentin microhardness and damage to the treated tooth [23].

Sodium hypochlorite (NaOCl) solutions are used as the main irrigation agent for removing the organic component of the smear layer because of their bactericidal power and capacity to dissolve organic matter and necrotic tissue [24]. NaOCl can physically flush out loose debris from root canals and serves as a lubricant for instrumentation [25]. In addition, it is an effective antimicrobial agent with the capability of detoxifying the root canal system and providing hemostasis in the bleeding pulp. Sodium hypochlorite irrigant is often used prior to or after EDTA to remove the inorganic components as well as dissolve the organic tissues. However, in certain final irrigation protocols, when NaOCl was used as the final flush after the irrigation of EDTA, the mechanical properties of dentin in the root canals may have been damaged by the proteolytic action of NaOCl [26,27,28]. More disadvantages of NaOCl in conjunction with EDTA as a root canal disinfectant have been discovered in several studies. The chemical interactions between chelating agents and NaOCl result in a loss of free available chlorine (FAC) and consequently decrease the tissue dissolution capability of NaOCl [29,30]. A novel continuous chelating root canal irrigant, 1 hydroxyethylidene-1, 1- bisphosphonate or etidronate (HEDP), was validated to maintain the proteolytic or antimicrobial properties of NaOCl [31]. Soft chelators such as HEDP have minimal effects on dentinal walls with weaker demineralization action compared to EDTA [32], but can simultaneously dissolve collagen and have a significant impact on the bond strength of epoxy-based sealer to dentin [33].

Two major factors are directly related to the efficiency of root canal irrigation during endodontic treatment: the delivery system used for irrigation and the properties of the irrigating solution. Traditional irrigating methods using syringes and needles have been proven insufficient to achieve complete cleanliness of the canal [34], especially in highly complex root canal systems with isthmuses and accessory canals where instruments cannot gain access [35]. Various devices for irrigation delivery and activation, including sonic and ultrasonic instruments, have been advocated to elevate the debridement efficacy of root canal systems [34,36]. Sonically driven irrigant activation systems comprised of flexible tips are not only designed to increase intracanal fluid flow, but also made to easily reach curved canals and the apical canal portion [36,37]. Passive ultrasonic irrigation (PUI), without simultaneous instrumentation, uses a stainless steel file to oscillate the irrigation solution and promotes cleanliness of the root canals [38]. However, due to the limited extension of PUI’s rigid metal tip, some studies have claimed that it is not effective for cleaning the apical area or severely curved root canals [39]. Both sonic and ultrasonic irrigation have better cleaning and disinfection effectiveness than syringe irrigation [40,41]. Nevertheless, whether sonic or ultrasonic is superior to other techniques remain controversial.

In the past, nanoparticles of silver [42], chitosan [43], iron oxides [44], and diamonds coated with amoxicillin [45] have been cited in bacteriostatic research. However, further investigations are necessary to determine whether nanoparticles can be applied to root canal cleaning. There is no evidence of a non-chelating irrigating solution that maximizes smear layer removal but does not concurrently diminish tissue dissolving properties of sodium hypochlorite. In addition, as mentioned above, the current irrigating solution is based on chemical reactions. There is no physical method to achieve the goal of smear layer removal, especially in the apical third of the root canal. Hence, the aim of the current study was to evaluate the effectiveness of nano and submicron diamond solution with sonic or ultrasonic oscillation on smear layer removal.

## 2. Materials and Methods

### 2.1. Instrumentation of Teeth Specimens

Eighty freshly extracted human single-rooted premolars were collected. Teeth referring to patients under eighteen years old, open apex, previously root treated, severe intracanal infection, and teeth with periapical lesions/symptoms were excluded. According to the standard procedures of non-surgical endodontic treatment, the pulp chambers were accessed with a round diamond bur in a high-speed handpiece, and preliminary shaping and recording of the working lengths were performed with #10 k-file (Dentsply Maillefer, Tulsa, OK, USA). The specimens were initially prepared using size 16, 0.02 taper 25 mm ProGlider^®^ Rotary Glide Path File and ProTaper^®^ Gold SX file (Dentsply Sirona Taiwan, Taipei, Taiwan) to expand the gliding path. Subsequently, four ProTaper^®^ Gold Ni-Ti rotary shaping and finishing files (Dentsply Sirona Taiwan, Taipei, Taiwan) were used at the working length in each canal according to the manufacturer’s instructions in the following sequence: (1) S1, size 18, 0.02 taper; (2) S2, size 20, 0.04 taper; (3) F1, size 20, 0.07 taper; and (4) F2, size 25, 0.08. After removing each rotary file, canals were irrigated with 5 mL of 3% NaOCl (Kojima Chemicals Co. Ltd., Saitama, Japan), and paper points were used to dry the canals after final preparation.

### 2.2. Preparation of Nano and Submicron Diamonds

The nanodiamonds used in this study are commercial products with a high-pressure, high-temperature synthesized process (HonWay Materials Co., Ltd., Kaohsiung, Taiwan). Five different sizes of nano and submicron diamonds were selected as test groups: 10, 50, 100, 500, and 1000 nm. The purity and particle size distributions from the grinding of diamond microcrystals were obtained and confirmed using a Particle Size Analyzer (LA-920, HORIBA, Fukuoka, Japan) (Figure 1).

### 2.3. Root Canal Irrigation Solutions

The prepared teeth specimens were divided into eight groups (*n* = 10) based on different irrigation solutions:Group 1 (Control): 3% NaOCl;Group 2 (Positive control): 17% EDTA;Group 3 (Negative control): distilled water;Group 4 (10 nm): 3% NaOCl and 10 nm nanodiamonds with a concentration of 10 mg/mL;Group 5 (50 nm): 3% NaOCl and 50 nm nanodiamonds with a concentration of 10 mg/mL;Group 6 (100 nm): 3% NaOCl and 100 nm nanodiamonds with a concentration of 10 mg/mL;Group 7 (500 nm): 3% NaOCl and 500 nm submicron diamonds with a concentration of 10 mg/mL;Group 8 (1000 nm): 3% NaOCl and 1000 nm submicron diamonds with a concentration of 10 mg/mL.

### 2.4. Sonic and Ultrasonic Vibration Oscillation

Irrigating solutions for each group were first delivered into the prepared teeth specimens with a 30-gauge syringe needle inserted 1 mm short of the working length without binding. Subsequently, sonic and ultrasonic instruments were used to oscillate 5 samples of each group separately for 10 s. Passive ultrasonic irrigation was performed with an ultrasonic instrument (Newtron P5, Satelec, Viry-Châtillon, France) equipped with a metal blunt working-end tip to activate the irrigants; the sonic-activated irrigation was applied with sonic instrument (EndoActivator, Dentsply Sirona, NC, USA) and its disposable polymer tip. Following oscillation, all specimens were irrigated with 3 mL of distilled water and dried with sterile paper points. All procedures were performed by the same operator.

### 2.5. Scanning Electron Microscopy (SEM) Analysis

The crown portion of each tooth specimen was cut off under a dental operating microscope (Extaro 300, Zeiss, Jena, Germany). Roots were sectioned perpendicularly to its long axis in equal proportions with a low-speed handpiece diamond disc into the coronal third, middle third, and apical third, respectively. After dehydrating and processing the root surface with gold palladium, the sectioned specimens were viewed under a scanning electron microscope (SU3500, Hitachi Ltd., Tokyo, Japan) in 500× and 2000× magnifications to examine the remaining smear layer. An evaluation was made to evaluate the smear layer removal effectiveness by recording the dentinal tubule exposure quantity as well as the presence, quantity, and distribution of the smear layer based on the score described by Hulsmann et al. [46]. The score was graded on a four-point scale as follows: score I = dentinal tubules completely opened, score II = more than 50% of dentinal tubules opened, score III = less than 50% of dentinal tubules opened, score IV = all of the dentinal tubules covered with smear layer. Two observers were calibrated, and a blind evaluation was performed by examining the images independently. The level of agreement between the two observers was assessed with the kappa test.

### 2.6. Statistical Analysis

The Kruskal–Wallis and Mann–Whitney U tests were used to evaluate the statistical difference between the measured results. Mean scores of the results recorded at the coronal, middle, and apical third were analyzed. Results with *p* < 0.05 were considered statistically significant.

## 3. Results

### 3.1. Examiner Calibration

Interrater reliability evaluated with the kappa test for scoring the smear layer removal showed a strong to an almost perfect level of agreement in each category. The kappa values are 0.94, 0.92, and 0.85 in the coronal, middle, and apical third of the root, respectively.

### 3.2. Smear Layer Removal of Irrigants under Sonic Oscillation

The use of 3% NaOCl (control group) alone was ineffective in removing the smear layer in all three sections of the root canal walls (Figure 2a). Under sonic agitation, various sizes of nano and submicron diamonds demonstrated equivalent efficacy in removing the smear layer as the positive control EDTA irrigant. Specimens irrigated with diamonds sized 50 nm, 100 nm, 500 nm, and 1000 nm were seen with dentinal tubules exposure in the SEM photographs (Figure 2a). Except for the 100 nm diamond solution, dentinal tubules of the other diamond solution groups were all equally patent in the coronal, middle, and apical portions of the root canal (Figure 2b). The root canals irrigated with diamond solutions showed rough wall surfaces with scattered fragments. The size of the impacted fragments on the root canal walls increased with the irrigation solution’s diamond size. Therefore, after irrigating with 1000 nm diamond solution and with the aid of sonic oscillation, larger fragments were scattered on the root canal wall surface (Figure 2b).

In the coronal and middle thirds of the specimens agitated by sonic instrument, there was no significant difference in the smear layer removal value among the 50 nm, 100 nm, 500 nm, and 1000 nm-sized diamond irrigation solution and the EDTA group (*p* < 0.05, Figure 3a,b). Furthermore, EDTA resulted in significantly higher scores of smear layer removal in the apical portion, which differed from the diamond sizes 50 nm and above irrigation solution groups (*p* < 0.05, Figure 3c). Moreover, among all the nano and submicron diamond irrigation solution groups, only the 10 nm-sized diamond group did not exhibit a significant difference when compared to the NaOCl and H_2_O groups.

### 3.3. Smear Layer Removal of Irrigants under Ultrasonic Oscillation

The use of ultrasonic oscillation showed that the groups using 500 nm and 1000 nm submicron diamond irrigation solution had significant smear layer removal (Figure 4a). In these two groups of diamond solutions that were effective in removing the smear layer, dentinal tubules in the coronal and middle thirds were significantly more exposed than those in the apical third (Figure 4b). In addition to being occluded, there was a decrease in tubular density in the apical third of the root canal wall. SEM evaluation indicated no trends in the canal wall pattern or morphology between each group of specimens after being irrigated by the diamond solution.

The evaluation of the smear layer of ultrasonically agitated specimens suggested that 500 nm and 1000 nm-sized submicron diamond irrigation solution significantly differed from the EDTA group in the coronal and middle section (*p* < 0.05, Figure 3d,e). However, in the apical third section of the root canal, EDTA had no statistically significant differences between NaOCl, H_2_O, and the diamond irrigation solution (*p* > 0.05, Figure 3f). In all three sections of the bisected root canals, 500 nm and 1000 nm-sized submicron diamond irrigation solutions showed significantly lower smear layer removal scores, while the groups of 10 nm, 50 nm, and 100 nm diamond irrigation solution did not significantly differ from the NaOCl, EDTA, and H_2_O groups (*p* > 0.05).

## 4. Discussion

Enlarging, shaping, and cleaning have long been recognized as essential procedures in root canal treatment. To eliminate the microbiologic etiology of pulpal disease, disinfection is always a key part of the treatment strategy. Besides the impressive innovation of various mechanical instruments, such as the use of super-elastic metal alloy files or the metallurgic improvements of rotary files, thorough and copious irrigation plays a pivotal role in supporting an overall debridement that also preserves sufficient root strength and a successful obturation outcome [40,47]. Some studies have reported that retaining the smear layer created by instrumentation can harbor bacteria and provide an avenue for leakage. The removal of the smear layer can benefit treatment outcomes [48]. Since the two most commonly used irrigation solutions, NaOCl and EDTA interfere with each other [49], an irrigation system that can continuously remove the smear layer during the process of cleaning and shaping is highly considered.

Smear layer components include very small particles with a large surface and are almost inevitable during root canal instrumentation. The most common chelating agents are based on EDTA, which acts upon the inorganic substances of the smear layer, causing the demineralization of peritubular and intertubular dentine. In spite of the fact that EDTA can demineralize dentine to a depth of 20–30 min in 5 min [50], it is well known that the chelating solution was rather ineffective in the apical third of the root canals. Moreover, Teixeira et al. (2005) [51] also stated that the smear layer was notably removed on the dentine surfaces of the root canal walls in coronal and middle thirds compared to the apical third after the irrigation of EDTA followed by NaOCl. This is because water molecules have strong, cohesive forces between them, known as surface tension. It is a property of liquids that arises from the cohesive forces between molecules at the surface of the liquid. When water flows through a thin tube, surface tension causes water to form a convex meniscus, creating a “water block” that hinders the free flow of water through the tube. Since the end of the root canal is very thin, the chelating solution cannot reach this area due to this water block phenomenon. According to Yamada et al. (1983) [16], comparable smear layer cleanliness from the coronal to the apical end of the root canal wall was achieved in their study; however, this result only existed when combined chemical activity of irrigants along with a direct high-volume flush after instrumentation has been completed. In comparison, the present study using nano and submicron diamonds to irrigate the root canals represented similar smear layer removal effectiveness in coronal, middle, and apical thirds, as shown in Figure 2 and Figure 3. It is possible that the mass of these diamond particles can reach the full length of the canal by ultrasound or hydrodynamic phenomenon. This is because nanoparticles, due to their small size and large surface area, can interact with the liquid at the nanoscale and potentially disrupt the surface tension of liquids, including water. Consequently, this allows diamond particles to impact the canal walls physically and result in the complete removal of the smear layer.

In a pilot study, we prepared acrylic models with dye on the canal walls to mimic the smear layer on root canal walls and irrigated them with a submicron diamond solution of different concentrations by sonic and ultrasonic oscillation. Our results showed that irrigation of 10 mg/mL submicron diamond solution was significantly effective. None of the other concentrations (1 mg/mL, 20 mg/mL, 50 mg/mL) were as effective as 10 mg/mL. In compliance with the preliminary results, our present study implies that effective cleaning action in all regions of the extracted teeth root canals, even the apical third, was obtained when irrigated with sonic and ultrasonic oscillation in 10 mg/mL of specific-sized nano and submicron diamond solution (Figure 2b and Figure 3b).

With the purpose of reaching favorable debridement in the apical portion of the root canal and overcoming the limitations of instrumentation in complex root canal anatomy, the development of various machine-assisted irrigation techniques or systems has gained increasing attention. Both sonic and ultrasonic activation, with the mechanism of acoustic streaming and hydrodynamic cavitation, are effective methods for disinfecting root canals [52,53,54]. Nevertheless, there are conflicting results about the performance of sonic and ultrasonic irrigation systems. According to Rodig et al. (2010) [55], the EndoActivator^®^ achieved significantly more smear-layer-free canal walls than ultrasonic agitation at the coronal level, but no difference was observed in the apical portion of the curved root canals. Other studies concluded that more powerful ultrasonic systems removed more dentin debris from the root canal than the less powerful sonic irrigation systems [56,57]. When comparing the smear layer removal efficacy between sonic and ultrasonic systems in activating various sizes of nano and submicron diamond solution, our result suggests that both systems can benefit the cleanliness of the root canal walls in all three sections, but either of the systems only takes effect when used with specific diamond sizes. As sonic agitation can provide 50 nm, 100 nm, 500 nm, and 1000 nm diamond solution to effectively remove the smear layer on the root canal walls, ultrasonic agitation can only allow rather larger diamond particles, which are 500 nm and 1000 nm diamond solution, to reach similar cleanliness. This can be explained by the kinetic energy equation: $E=\frac{1}{2}mV^{2}$. When applying nano and submicron diamonds in the irrigation system, the mechanism of action for activating these diamond particles through sonic and ultrasonic systems are particularly different. Sonic irrigation operates at a lower frequency (160–190 Hz) and generates a higher amplitude that is believed to be capable of maximizing hydrodynamics and three-dimensional disinfection [58]. By automatically vibrating the tip and manually making short vertical stroke movements of the tip, fluid activation, and intracanal waves are generated. The waves conduct energy directly to diamond particles and subsequently hit canal walls to remove smear layers physically. On the contrary, ultrasonic technology produces high-frequency (25–30 kHz) sinusoidal waves with low amplitude [59]. During passive ultrasonic irrigation, the energy is transmitted from an oscillating file to the irrigant in the root canal by means of ultrasonic waves. The latter induce acoustic streaming and cavitation of the irrigant. High-frequency vibration produces small bubbles when oscillating the solution, and the small bubbles agitate diamond particles in the solution. As the disturbance accelerates, small bubbles undergo collapse and rebound. Finally, in the succession of collapse and rebound cycles, fragmentation and disappearance of the bubbles caused by a single microjet or a spherical harmonic disturbance result in the termination of energy transfer [60]. This is the possible reason why ultrasonic instrument is ineffective in smaller particle sizes of nanodiamonds.

When comparing the characteristics of sonic and ultrasonic devices, a major difference is the tip material of these two. The EndoActivator^®^ System is equipped with disposable polymer tips of different sizes. These tips are claimed to be strong, smooth, and flexible that do not break easily and cut dentin [34]. The EndoActivator^®^ System was reported to be able to effectively clean debris from lateral canals, remove the smear layer, and dislodge clumps of simulated biofilm within the curved canals, as well as maintain the anatomical integrity of the final preparation [61]. In contrast, ultrasonically driven instruments are manufactured from metal alloys. Even with pre-curved ultrasonic tips, there are still clear constraints and restrictions when applying these tools in curved canals and lengthening [34]. There is always a chance of contacting the dentin walls, which would immediately result in the formation of a new undesirable smear layer. Moreover, applying a rigid vibrating tip around a canal curvature invites iatrogenic events, including ledges, apical transportation, lateral perforations, or broken instruments [62]. In the present study, when using straight root canals as models, there was no significant difference observed between sonic and ultrasonic agitation in terms of the effect of smear layer removal. The difference between sonic or ultrasonic oscillation of nano and submicron diamond solution in curved canals can be examined in greater detail.

The results of this investigation demonstrated the improvement of smear layer removal in the apical third of root canals, indicating that nano and submicron diamond solution can be an alternative irrigant for root canal treatment; however, the success of treatment does not only lie in the elimination of smear layer. Research in the effect of submicron and nano diamond on biofilm removal, antibacterial ability, and sealer penetration requires to be further investigated.

One limitation of this study is the lack of longitudinal observational character, in which a given dentin area cannot be observed at different times due to the sample dissection and preparation for SEM evaluation [63]. However, an ideal experimental model to assess smear layer removal is not currently available. Further research efforts can be focused on designing a more reliable and reproducible smear layer removal experiment. The other limitation is that the time of EDTA staying in contact with the root canal walls is set in consistence with the nano and submicron diamond solutions in the present study, as this time setting was regarded effective in the removal of smear layers for nano and submicron diamond solution. However, some protocols suggest two or three cycles of 20~30 s oscillation for chelating agents to act [64]. Future studies can also compare the smear layer removal effectiveness of chelating agents and nano and submicron diamond solutions when simulating the oscillation time of clinical endodontic procedures.

## 5. Conclusions

Sonic and ultrasonic agitation of specific sizes of nano and submicron diamond solution both promoted the smear layer removal, while the diamond solution showed greater efficacy in the apical third when compared with EDTA irrigation.

## Figures and Tables

**Figure 1 nanomaterials-13-01646-f001:**
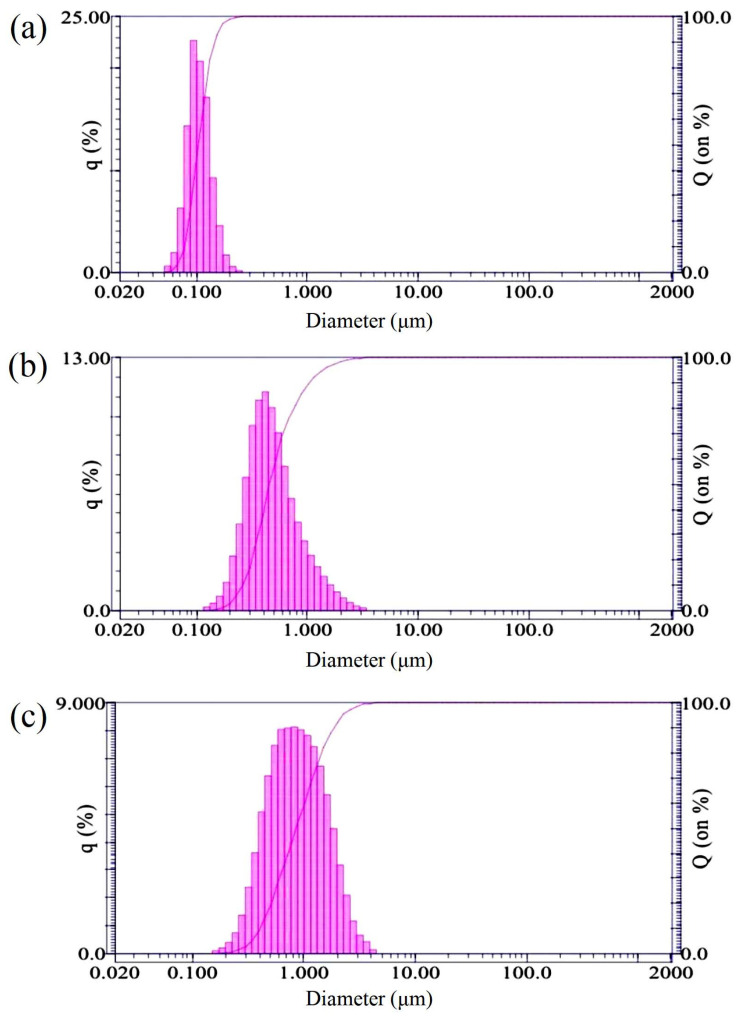
Particle diameter distributions of (**a**) 100 nm, (**b**) 500 nm, and (**c**) 1000 nm nano-diamond particles.

**Figure 2 nanomaterials-13-01646-f002:**
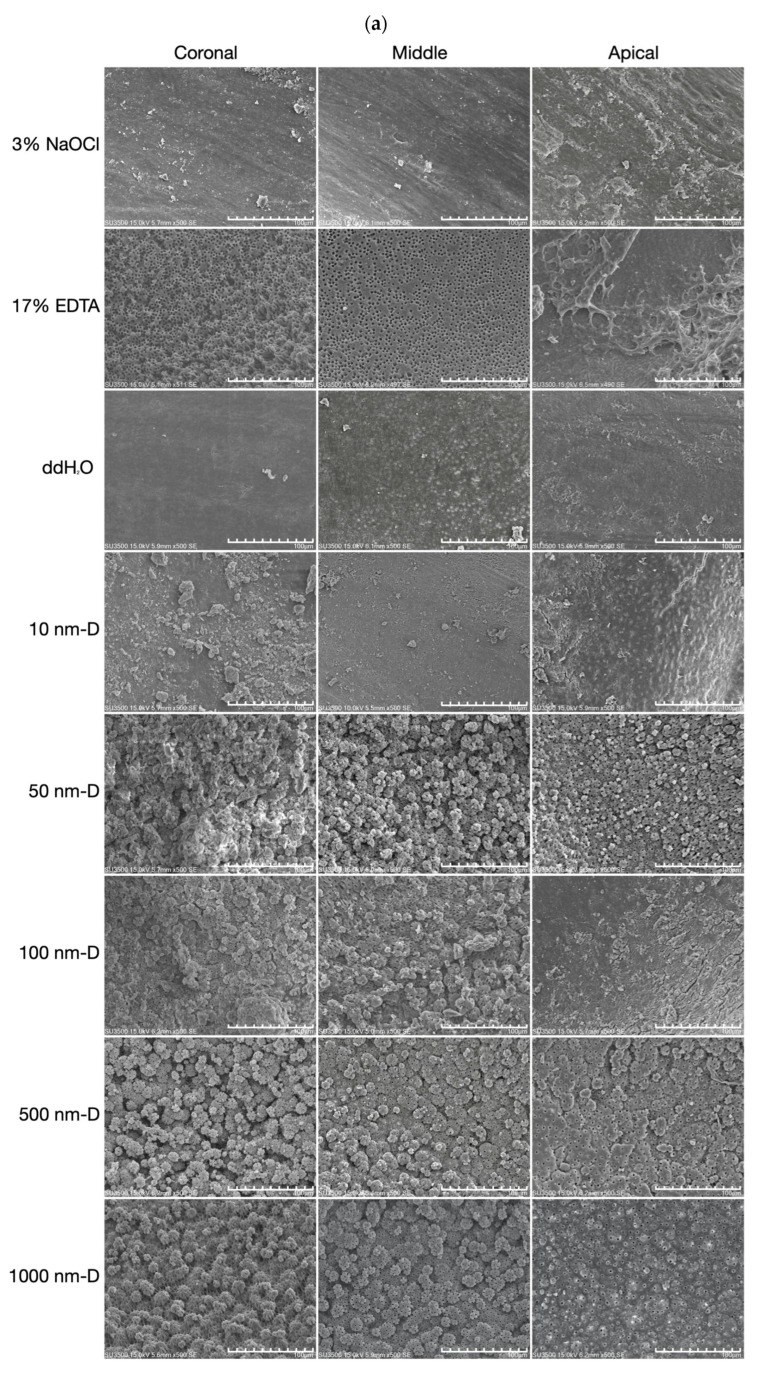
Scanning electron microscopy photomicrographs of the root canal walls after sonic agitation: (**a**) Micrographs showing all control and test groups’ dentin surfaces in the coronal, middle, and apical portions under the magnification of 500×. The white bar represents 100 μm. Dentinal tubule exposure is not obvious in control (3% NaOCl) and negative control (distilled water) groups, where large surfaces of smear layer covered the dentinal tubules indicating poor smear layer removal effectiveness. (**b**) Dentinal tubules (marked with arrows) exposed in the groups of 50, 100, 500, 1000 nm nano and submicron diamond irrigation solution under the magnification of 2000×. The white bar represents 20 μm. In the apical thirds of the specimens agitated by sonic instrument, there was significant difference in the smear layer removal among the 50 nm, 100 nm, 500 nm, and 1000 nm-sized diamond irrigation solution and the EDTA group.

**Figure 3 nanomaterials-13-01646-f003:**
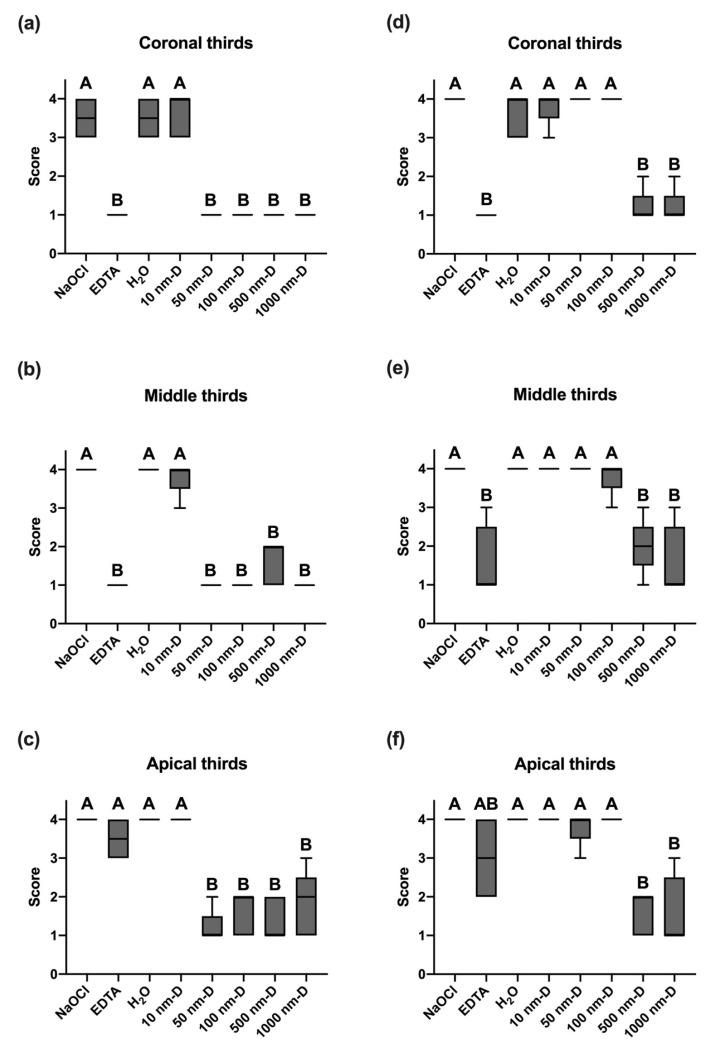
Box plots display the mean, minimum, and maximum values, as well as the variance of the smear layer evaluation score in all groups: (**a**) Sonic-activated irrigation at coronal third. (**b**) Sonic-activated irrigation at middle third. (**c**) Scores of groups with sonic-activated irrigation at apical third. (**d**) Ultrasonic-activated irrigation at coronal third. (**e**) Ultrasonic-activated irrigation at middle third. (**f**) Ultrasonic-activated irrigation at apical third. Box plots of the six groups’ data show the median, standard deviation, and variance in each group. Different letters indicate statistically significant differences; *p* < 0.05 was considered significantly different.

**Figure 4 nanomaterials-13-01646-f004:**
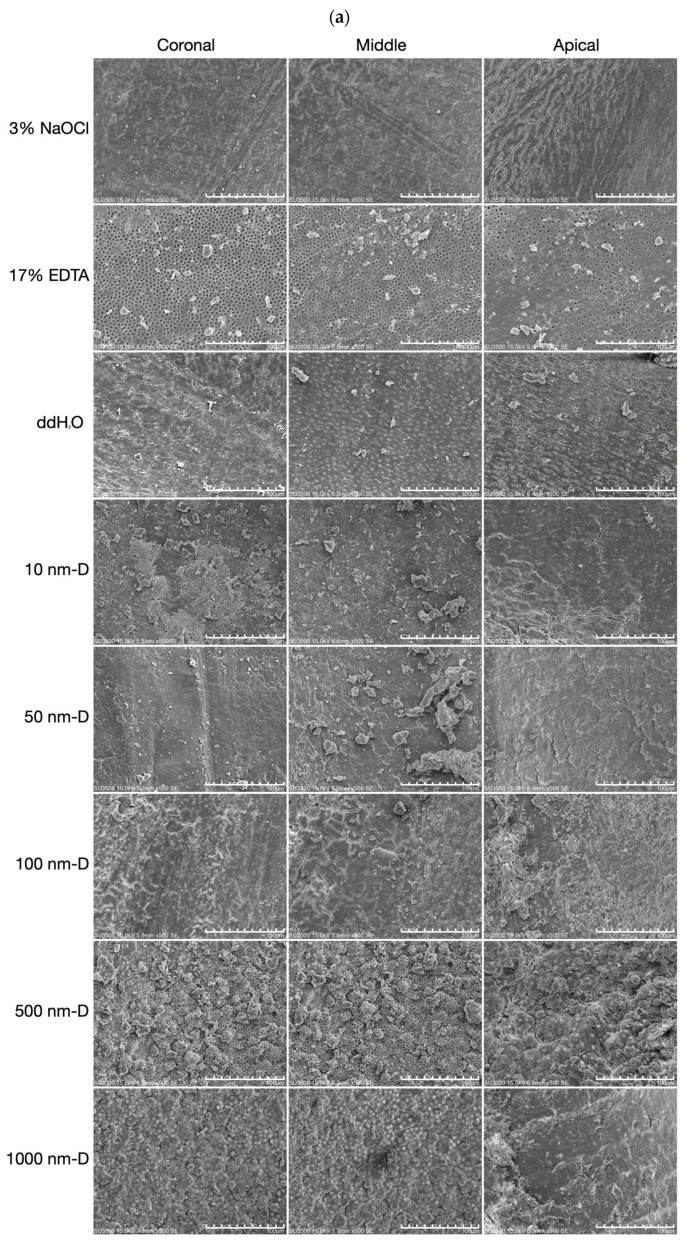
Scanning electron microscopy photomicrographs of the root canal walls after ultrasonic agitation: (**a**) Micrographs showing all control and test groups’ dentin surfaces in the coronal, middle, and apical portions under the magnification of 500×, and the white bar represents 100 μm. (**b**) Dentinal tubules (marked with arrows) exposed in the groups of 500 and 1000 nm submicron diamond irrigation solution under the magnification of 2000×. The white bar represents 20 μm. Smear layer removal effectiveness of 500 nm and 1000 nm-sized submicron diamond irrigation solution was comparable to EDTA in the coronal and middle section but not as effective in the apical third of the root canals.

## Data Availability

Data is unavailable due to privacy or ethical restrictions.

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
