# Peer review of "A Novel Endodontic Approach in Removing Smear Layer Using Nano and Submicron Diamonds with Intracanal Oscillation Irrigation"

_nanomaterials, 2023, doi:10.3390/nano13101646_

Round 1
Reviewer 1 Report
To work presents the efficacy of applying nano and submicron diamonds in irrigation solutions with sonic and ultrasonic oscillation for removing the smear layer on root canal walls. While I find the topic interesting, there are several major issues that need to be addressed before this manuscript can be considered for publication.
Firstly, the manuscript appears to be more relevant to the dentistry field rather than the field of nanomaterials. The authors should clarify the relevance of the use of nanodiamonds in this study and how it fits into the broader context of nanomaterials research.
Secondly, the authors should provide a more detailed discussion of the mechanisms by which NaOCl and EDTA affect the smear layer.
Thirdly, the authors should explain in more detail how and why nanoscale diamonds have a better effect in protecting the root. The authors should provide a thorough explanation of the physical and chemical properties of nanodiamonds that make them more effective for this purpose.
Finally, the SEM image in Figure 1 lacks a distance scale, and the differences between the diamond-treated samples and the control samples are not immediately apparent. The authors should provide more detailed and comprehensive SEM images that allow the reader to clearly see the differences between the different samples.
In summary, while the topic of this manuscript is interesting, it requires major revisions before it can be considered for publication. The authors should address the issues mentioned above and provide more comprehensive and detailed explanations of the underlying mechanisms and the effectiveness of nanodiamonds in protecting the root.
The English is OK in generall, however the author should use more "vocabulary" in nano and material science, not in densitry.
Author Response
Point 1: Firstly, the manuscript appears to be more relevant to the dentistry field rather than the field of nanomaterials. The authors should clarify the relevance of the use of nanodiamonds in this study and how it fits into the broader context of nanomaterials research.
Response 1: We thank the reviewer for these comments. In the revised manuscript, the relevance of the use of nanodiamonds in this study was added on page 5, last paragraph.
Point 2: Secondly, the authors should provide a more detailed discussion of the mechanisms by which NaOCl and EDTA affect the smear layer.
Response 2: Thank you for the comment. Detailed discussion in regard to the effect of NaOCl and EDTA on smear layer is added in page 3 second and third paragraph of the introduction section.
Point 3: Thirdly, the authors should explain in more detail how and why nanoscale diamonds have a better effect in protecting the root. The authors should provide a thorough explanation of the physical and chemical properties of nanodiamonds that make them more effective for this purpose.
Response 3: We thank the reviewer for these comments. Since nanomaterials Nanoparticles, due to their small size and large surface area, can interact with the liquid at the nanoscale and potentially influence the surface tension of liquid, including water. It can not only can provice the source or impact force on root canals, but also help to disrupt the water block phenomenon and make the irrigation solutions can flow to the whole root canal. We added these description to the last paragraph of page 11 and the first paragraph of page 12.
Point 4: Finally, the SEM image in Figure 1 lacks a distance scale, and the differences between the diamond-treated samples and the control samples are not immediately apparent. The authors should provide more detailed and comprehensive SEM images that allow the reader to clearly see the differences between the different samples.
Response 4: Thank you for the comment. Figure 1 and 2 has been revised and distance scale is added on every SEM image, and the figure legend has been edited to explain about the smear layer removal effectiveness. Since the coronal and middle portion of diamond-treated groups are not significantly different to the positive control (17% EDTA), SEM images may not look much different from each other. Dentinal tubule exposure is not obvious in control (3% NaOCl) and negative control (distilled water) groups, which indicates poor smear layer removal effectiveness.

Reviewer 2 Report
In the current manuscript, the authors suggest that sonic and ultrasonic oscillation with specific sizes of nano and submicron diamonds irrigation solution can be used as an alternative approach to remove smear layer.
Unfortunately, I didn't see any experimental measurement about nano level. This is the main absence of the manuscript. I suggest to present and analyze the high-pressure-high-temperature synthesized nanodiamonds, and demonstrate the size (10, 50, 100, 500, 1000 nm). Could be mentioned if these particles remain at the processed sample or not, and if these could be pointed on images.
Also, just some SEM images who is also without scale bar and without interpretation is not enought for the quality of this journal. It is mandatory to process the SEM images and point the interested issues (like "Smear layer components include very small particles with a large surface") on each images. Authors didn't explain to the readers how was evaluated “the smear layer removal effectiveness”.
According to the figure 2, the diameter of the dentinal tubules looks to be different after each treatment. I suggest to follow this issue and perform some measurements and discussion on this issue.
Level of the references could be improved.
Author Response
Point 1: Unfortunately, I didn't see any experimental measurement about nano level. This is the main absence of the manuscript. I suggest to present and analyze the high-pressure-high-temperature synthesized nanodiamonds, and demonstrate the size (10, 50, 100, 500, 1000 nm). Could be mentioned if these particles remain at the processed sample or not, and if these could be pointed on images.
Response 1: We thank the reviewer for these comments. The nanodiamond used in this study was a commercial product. We did check the diameter of these diamonds with Particle Size Analyzer (LA-920, HORIBA, Fukuoka, Japan). The following figures demonstrated the size distribution of the used nanodiamond with a diameter of 50, 100, 500 and 1000 nm. The tests confirmed that the particle sizes of used nanodiamonds were within their commercial specification. As the comments from the reviewer, we added a sentence to mention this confirmation.

Point 2: Also, just some SEM images who is also without scale bar and without interpretation is not enought for the quality of this journal. It is mandatory to process the SEM images and point the interested issues (like "Smear layer components include very small particles with a large surface") on each images. Authors didn't explain to the readers how was evaluated “the smear layer removal effectiveness”.
Response 2: Thank you for the comment. Figure 1 and 2 has been revised and scale bar is added on every SEM image. The figure legend has been edited to explain about the smear layer removal effectiveness. An evaluation was made to evaluate the smear layer removal effectiveness by recording the dentinal tubule exposure quantity as well as the presence, quantity, and distribution of smear layer based on the score described by Hulsmann et al. The above mentioned has been supplemented in page 8 line 2 and 3 of the materials and methods section.
Point 3: According to the figure 2, the diameter of the dentinal tubules looks to be different after each treatment. I suggest to follow this issue and perform some measurements and discussion on this issue.
Response 3: Thank you for the comment. The diameter of dentinal tubules may vary between each teeth and each canal portions. Apical third of the root has the lowest dentinal tubules density and smallest diameter, while coronal third has the highest density and greatest diameter [1-3].
- Carrigan, P.J.; Morse, D.R.; Furst, M.L.; Sinai, I.H. A scanning electron microscopic evaluation of human dentinal tubules according to age and location. Journal of Endodontics 1984, 10, 359-363, doi:https://doi.org/10.1016/S0099-2399(84)80155-7.
- Mjör, I.A.; Nordahl, I. The density and branching of dentinal tubules in human teeth. Archives of Oral Biology 1996, 41, 401-412, doi:https://doi.org/10.1016/0003-9969(96)00008-8.
- Lo Giudice, G.; Cutroneo, G.; Centofanti, A.; Artemisia, A.; Bramanti, E.; Militi, A.; Rizzo, G.; Favaloro, A.; Irrera, A.; Lo Giudice, R.; et al. Dentin Morphology of Root Canal Surface: A Quantitative Evaluation Based on a Scanning Electronic Microscopy Study. Biomed Res Int 2015, 2015, 164065, doi:10.1155/2015/164065.

Round 2
Reviewer 1 Report
The MS is pubishable after the revision.
There is still several typo in the revised MS. A language check should be done before publishing.
Author Response
Thank you for your comment. We have edited the grammar and typo problems in the revised MS.